# An Inductive Discussion of the Interrelationships between Nursing Shortage, Horizontal Violence, Generational Diversity, and Healthy Work Environments

**Francesca Armmer**

Department of Nursing, Bradley University, Peoria, IL 61625, USA; faa@fsmail.bradley.edu

**Abstract:** The complex features of the nursing shortage, horizontal violence, generational diversity and healthy work environments have frequently been addressed within the context of their singular characteristics, challenges and potential solutions. Yet it is the interrelationships of these phenomena that holds solutions to the overarching challenges facing nurses and the nursing profession. Through an inductive approach, a preliminary discussion and related strategies to address the highlighted challenges have been proposed.

**Keywords:** nursing shortage; horizontal violence; generational diversity; healthy work environments

---

## 1. Introduction

Complex features of the nursing shortage, horizontal violence, generational diversity, and healthy work environments have oftentimes been addressed within discussions that fostered an individual examination of the respective challenges. Yet it is the interactive impact of each of these components that continues to influence the nursing profession.

It is critical that an ongoing examination of factors that impact the nursing shortage, and ultimately the provision of quality patient care, be explored. A preliminary inductive discussion is offered in this paper based upon the following premises:

- There is a current and continuing nursing shortage.
- Horizontal violence is one phenomenon that influences nurse retention and thereby the nursing shortage.
- Nurses who experience horizontal violence are faced with a decisional response regarding intent to remain in the work setting.
- Generational diversity may be the stimulus as well as the adverse outcome of horizontal violence.
- Healthcare and nursing administrators have an opportunity to foster strategies to address horizontal violence.
- Countering horizontal violence will foster the creation of healthy work environments and thereby contribute to increased nurse retention and reduced nursing shortage.

## 2. Nursing Shortage

In May 2014, registered nurses held approximately 2.8 million jobs, with 61% of the employment in hospitals, either state, local, or private (U.S. Bureau of Labor Statistics 2015a). Employment opportunities and salary contribute to the appeal of the nursing profession. Bureau of Labor Statistics data has reported that in May 2015 the annual median wage for registered nurses was $67,490, with the lowest ten percent of nurses earning less than $46,360 and the highest ten percent earning more

than $101,630 (U.S. Bureau of Labor Statistics 2015b). Yet workforce shortages persisted. The American Association of Colleges of Nursing has reported that new positions that have been created for registered nurses are not being filled (American Association of Colleges of Nursing 2013). It is projected that more than 581,000 new positions will be available through 2018 (American Association of Colleges of Nursing 2013). Reports from the United States Bureau of Labor Statistics have projected the employment need for nurses in 2024 to be 3,190,000 (U.S. Bureau of Labor Statistics 2015c). Based upon this projection, the demand for registered nurses will continue in the next several decades.

Additionally, the challenge of unsuccessful retention of nurses has a tremendous impact upon quality of care to populations who enter the health care system with expectations of receiving optimal care. The approximate cost to replace a nurse who has practiced in the medical-surgical area is $92,444. To replace a nurse who has practiced in a specialty area will cost approximately $145,000 (Huddleston and Gray 2016a). Consequently, the nursing shortage and the associated financial impact of unsuccessful nurse retention have both immediate effects as well as long-term effects.

The International Council of Nurses has reported the pervasive nature of the nursing shortage via data that high-income countries maintain a nurse-population ratio almost eight times greater than low-income countries. This information translates to a range of less than 10 nurses per 100,000 persons to more than 1000 nurses per 100,000 persons (International Council of Nurses 2005). A sustained awareness of the nursing shortage is crucial. A discussion of horizontal violence as a plausible contributor to the nursing shortage is worthwhile.

## 3. Horizontal Violence

Results from the 2011 Health and Safety Survey of the American Nurses Association indicated that one in 10 nurses had been physically assaulted in the past year, and one third of the respondents identified one of the top safety concerns as on-the-job assault (ANA 2012). Horizontal violence, one type of workplace violence, has been described as "intergroup conflict that is manifested in overt and covert non-physical hostility such as sabotaging, infighting, scapegoating and bickering" (Duffy 1995). Bullying is a term that is also used for horizontal violence (McAvoy and Murtagh 2003). A study of perceptions of horizontal violence and the related responses from registered nurses continues to be studied (Armmer and Ball 2015). Based upon a random sample of 300 nurses, from which 104 useable response sets were analyzed, data that is directly critical in addressing the nursing shortage was obtained. As the researchers examined the presence of horizontal violence, a specific response to the phenomenon was explored, intent to leave. It was observed that, in response to horizontal violence experiences, younger nurses were more willing to leave their positions than were older nurses. It was also interesting that, while 51.9% of the respondents were not thinking about leaving their current positions, there was an expression of confidence that if a decision was made to leave, another position could be readily obtained (Armmer and Ball 2015). A subtle expression of personal power was communicated in the understanding that if an intent to leave decision were made, seeking employment would not be viewed as a difficulty. In a mixed methods study conducted by Moore et al., a sample of 400 nurses was asked to participate in a study examining nurse-to-nurse relationships and their impact on work environments. A response rate of 21% was reported. Researchers shared that 41 (55%) of the participants responded that they had left or considered the option of leaving their workplaces. When further questioned, participants identified three explanations for their considerations. These explanations were "frequent unit conflicts, presence of cliques, and lack of managerial support" (Moore et al. 2013, p. 175).

The prevalence of horizontal violence warrants an examination of experiences from a viewpoint of common occurrence rather than from the viewpoint of an exceptional or rare occurrence. Based upon a horizontal violence common occurrence viewpoint, a discussion of intent to leave and the impact of the nursing shortage becomes vital. Professional nurses are clear as to what they personally perceive to be an experience of horizontal violence. Professional nurses frequently experience horizontal violence

within their practice settings. Professional nurses of various age groups and various practice settings experience horizontal violence. The pervasive nature of horizontal violence must be addressed.

For the health care administration team or nurse administrator, the impact of these results is daunting. Regardless of measures to ameliorate or disavow a horizontal violence experience, the nurse(s) involved will hold a personally clear understanding of what has been experienced. Regardless of the absence of a focused policy or guideline for consistently addressing the experience, professional nurses are acutely aware that they may experience horizontal violence regardless of their age and regardless of their years of professional experience. The presence of horizontal violence, in any form, presents an adverse counter effect to strategies designed to focus upon retention of professional nurses and recruitment into the nursing profession (Becher and Visovsky 2012).

Consequently, an administrator will need to intentionally plan to address the challenges enmeshed in each horizontal violence experience. It will be the efficacy and compassion of the leader/administrator that will be incorporated into the nurse decision regarding intent to leave or not to leave. The consistency and effectiveness of a developed program of interventions will contribute to the survival and reputation of the institution(s) within the health care system.

## 4. Generational Diversity

It is essential that health care and nurse administrators consider the major construct of generational diversity as a potential contributing factor to horizontal violence. It is the perspective of generational diversity that provides insight into decisions associated with the professional nurse's intent to leave or not to leave. Outcomes from individual decisions made by each professional nurse will culminate in nurse retention or nursing shortage.

In an effort to describe and hopefully understand the nursing workforce, categories related to age have been utilized. For example, Baby Boomers have a birth date range of 1941–1964; Generation Xers have a birth date range of 1960–1981, and Millennials (Generation Y) have a birth date range of 1980–1999 (Apostolidis and Polifroni 2006). Baby Boomers have been characterized as nurses who are committed to the place where they work; who value promotions, titles and recognitions (Apostolidis and Polifroni 2006). Generation X nurses are characterized as viewing a job as temporary; are motivated by continuing education, training and income. Loyalty and pension plans are not motivating factors for the Generation X nurse (Apostolidis and Polifroni 2006). Apostolidis and Polifroni (2006) cited that "31% of nurses younger than 24 years will change jobs within the first two years of employment" (Apostolidis and Polifroni 2006, p. 507). The nurse who is in the Generation X category prefers to be mentored. Nurses within the Millennial generation are characterized by multi-tasking abilities, by an attitude of acceptance of technology as commonplace, and have never experienced a time without "touch-tone telephones, microwave ovens, videocassette recorders, and personal computers" (Olson 2009, p. 11). According to Lipscomb (2010), Millennials (Generation Y) are "loyal to a company that provides them with the best schedule, most money and the latest electronical gadgets. Generation Y'ers are multi-taskers and likely to have more than one job at the same time. They may be slow to accept additional responsibility. Balance is the Generation Y mantra" (Lipscomb 2010, p. 267).

When understanding workplace behavior, Brunetto et al. have raised an awareness that descriptions of generational cohorts may be too general to be used alone (Brunetto et al. 2013). Generational diversity as described by the characteristics of the cohorts may be valuable for general applicability with consideration of differences based upon the individual nature of each nurse. The characteristics of each cohort partnered with behaviors from the individual nurse create an intrapersonal dyad. This dyad of behaviors may result in experiences that fall within the category of horizontal violence (Becher and Visovsky 2012). The administrator is challenged to develop a strategic plan that will yield a positive response from the nurse to the "intent" decision. It is this "intent" decision that is faced by each nurse who experiences horizontal violence.

A foundation for the development of strategies to address generational diversity has been identified via the establishment of a "framework of values, beliefs and work ethics of each

generation cohort" (Lipscomb 2010, p. 269; Hendricks and Cope 2012). Through a process of understanding, team members' judgements and preconceived erroneous ideas may be lessened. Thus, through this communication framework, incidences of horizontal violence may be reduced. Participation in establishing a framework of values as an activity is reinforced by evidence presented by Apostolidis and Polifroni (2006). They reported that, in a study of importance and work satisfaction, both Baby Boomers and Generation Xers identified the same top three components: autonomy, professional status, and interaction (Apostolidis and Polifroni 2006). With the establishment of common understandings, discussions between cohorts become more readily facilitated. As a framework for ongoing understanding evolves a healthy work environment may be established and may become ultimately sustainable.

Complementary to the establishment of a framework of values is the use of reciprocal learning. Strom and Strom (2015) described reciprocal learning as "mutual growth based upon consideration of feelings, ideas, values, and perspectives of a different generation" (Strom and Strom 2015, p. 45). Preceptors, first line managers, supervisors, administrators and executive officers will need to embrace the experience of reciprocal learning through role modeling processes of valuing each nurse and the contributions that are made in the provision of quality patient care. Olson (2009) affirmed the importance of interpersonal behavior in the healthcare setting. Participants placed more importance on relationships and a respectful environment as facilitating their overall orientation experience than either the content of the orientation plan or the length of the orientation (Olson 2009). The tone of relationships and respect is set via leadership.

As the nurse-to-nurse and nurse-to-supervisor relationships are strengthened via diverse activities aimed at achieving reciprocal learning, outcomes of professionalism and collegiality are fostered. A healthy working environment will be sustained. Bullying is minimized. Retention enhanced. Recruitment strengthened.

## 5. Healthy Work Environment

Healthy Work Environments have been described as settings whose infrastructures of policies, procedures, and processes have been developed to empower nurses to maintain personal satisfaction and achieve organizational objectives (Huddleston and Gray 2016a, 2016b). The American Association of Critical Care Nurses has identified eight standards for a healthy working environment. These standards are authentic leadership, skilled communication, true collaboration, appropriate staffing, meaningful recognition, effective decision making, genuine teamwork, and physical and psychological safety. In contrast, unhealthy work environments are characterized by abusive behavior, disrespect, lack of trust, rigidity to change, poor communication and weak leadership with no vision (Huddleston and Gray 2016a). Incidences of horizontal violence are indicative of an unhealthy work environment. The establishment of a framework of values through the use of reciprocal learning may offer the initial steps in the cultivation of a healthy work environment and address the immediacy of nurse needs. Consideration of architecture and physical environment design may further contribute to a comprehensive strategic plan.

The aspect of aging is a compelling feature of generational diversity that should not be overlooked. In 2013 the average age of the registered nurse was 46.8 years. Within the profession of nursing, nurses aged 50–69 years are the largest cohort that has ever been represented in the United States of America (Stichler 2013). This aging nurse population presents another opportunity for nurse administrators to strategically explore workplace design initiatives, scheduling initiatives, and team initiatives that may result in positive retention approaches and health work environments. The cumulative impact of measures to foster a healthy working environment for all, including the aging nurse, incorporates an additional descriptor to the term healthy work environment. The descriptor is a "healing environment" (Trossman 2014). A healing environment reflects "design features that promote a healthy safe working environment for all health care professionals" (Trossman 2016, p. 959). Through the visionary leadership of nursing and health care administrators,

structural environmental challenges/barriers may be proactively addressed; thus maintaining a healthy, healing work environment.

The confounding presence of challenges to nurse retention and recruitment are aggravated by horizontal violence. And the resultant decisions of professional nurses whether to remain at the institution or leave following exposure to horizontal violence are influenced by the ongoing nature of a healthy work environment.

Experiences associated with horizontal violence have a dual impact. This impact is initially felt by the persons involved and from a residual perspective the impact is cumulative as the nurse relives the previous interaction(s) when other comparable experiences occur. Horizontal violence is a destructive distraction to the practicing nurse. Optimal patient-centered safe care is replaced as priority one when a range of bullying experiences are displayed in the work place. The aim of fostering a healthy work environment is for the client to be provided comprehensive, safe, thorough, compassionate care; and for health care providers to work in an environment that fosters respect and integrity, and provides for professional growth.

## 6. Strategies and Conclusions

The nursing profession continues to be faced with a major challenge. The continued nursing shortage is that challenge. In this discussion paper, interrelationships between horizontal violence, generational diversity and the nursing shortage have been introduced. Preliminary research evidence associated with nurse perceptions of horizontal violence affirmed that professional nurses are clear in their perceptions of horizontal violence, horizontal violence is a common experience, and that younger nurses reported a greater intent to leave the facility after horizontal violence experiences. Professional nurses who were currently not intending to leave the institution expressed a confidence that they could leave and find another nursing position, if necessary.

The decisional outcome by the nurse is an intent to leave or not to leave. This decision clearly effects the nursing shortage.

Strategic planning efforts that may result in actions and interventions that are focused upon influencing the nurse's intent have been explored. Strategies that foster a focus upon the generational diversity within the profession included the cultivation of a framework of values and reciprocal learning. Measures that foster a healthy and healing work environment have been presented. It will be important for the nursing and health administration leaders to maintain a dynamic vision for nursing that acknowledges the devastating effect of horizontal violence and the power of a healthy healing environment that celebrates the generational diversity of the profession. This vision may be operationally strengthened by activities that include an ongoing review and revision of the orientation program for new nurses, as well as refresher programs for current health care staff. Collaborating with area nursing education programs in the cultivation of a "real-time" understanding of the upcoming future employment experience will provide another venue for strengthening nurse recruitment and retention and decreasing potential episodes of horizontal violence.

There will be a multifunctionality component associated with these and other strategies[1] (Brunetto et al. 2013; Shermont and Krepcio 2006). This component will be reflected in the intentional efforts of leaders and administrators to establish teams of professionals from diverse backgrounds within the health care setting who will collaborate in the creative accomplishment of fostering a healthy, healing environment. Addressing the intersecting characteristics of nursing shortage, horizontal violence, generational diversity, and healthy work environments phenomenon is critical. The outcome will be increased nurse retention, increased response to nurse recruitment efforts, and sustained quality of patient care.

---

[1]  Business Dictionary. Available online: http://www.businessdictionary.com/definition/multifunctional-team.html (accessed on 20 September 2017).

**Acknowledgments:** No funds were received to cover support of this article development.

**Conflicts of Interest:** The author declares no conflict of interest.

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
