# Peer review of "An Inductive Discussion of the Interrelationships between Nursing Shortage, Horizontal Violence, Generational Diversity, and Healthy Work Environments"

_admsci, doi:10.3390/admsci7040034_

Round 1

Reviewer 1 Report

I thoroughly enjoyed reading this manuscript and agree with your proposed strategies. No doubt in my mind that the nursing shortage is tied to a healthy workplace environment or the lack there of. My only suggestion is that you re-read for very minor edits - for example line 54 would better read, " The approximate cost to replace a nurse....." or "The approximate cost of replacing a nurse..."

Author Response

Thank you for the feedback and encouragement.  I have made the editorial changes that you recommended, as well as the grammatical changes needed. 

Reviewer 2 Report

The additions to the paper that the authors have made are useful to help explain the intent of the paper.  Below are questions and suggested edits that are intended to help bolster the argument you are posing in the paper and add clarity to ideas presented.

Suggested edits:

-title – the order of the concepts listed in the title should align with the how these concepts are listed in the paper i.e. nursing shortage, horizontal violence, etc.

-p.1

-line 15- replace the comma with a period to end the sentence. Place a comma after ‘Yet’

-line 28 – “This paper will…” is anthropomorphic i.e. a paper is an inanimate object and cannot have an action verb. Suggest revising to a preliminary inductive discussion is offered in this paper based upon the following premises”

-lines 30-40 – This additional discussion you have inserted helps to clarify the central argument for this paper. I recommend that each bullet point is made into sentence format (better form).  As well, these statements need to be supported with cited literature.

-p. 2

-line 50 – the data provided is focused on the U.S., suggest considering adding in data from a more global perspective to reflect the broader audience for this journal.

-line 59 – suggest adding in a linking statement to bridge the discussion between nursing shortage and horizontal violence (e.g. after the first sentence in the paragraph).

-line 64 to 68–To be more explicit and succinct, suggest revising these statements as follows, “In a more recent study of 104 nurses, Armmer and Ball found that…(insert results rather than inserting a figure from their paper)”. The link to the nursing shortage needs to be clear.

-lines 71-74 – These statements can be integrated to be more succinct e.g. “ Given that nurses from various age groups and practice settings frequently experience horizontal violence, the pervasive problem of horizontal violence must be addressed.”

-lines 76-80 – No new ideas are presented here in these sentences, suggest deleting.

-lines 83-90 – This paragraph would be better placed after  line 74.  The discussion about the role of nursing administrators can be follow. It seems as though much of this discussion is based on a single study with a small sample size, suggest locating additional studies published since 2011 to strengthen your argument.  

-p.3

-line 96 – since generational diversity is a new idea, begin a new paragraph for this discussion. Generational diversity as one of the causes of horizontal violence needs to be supported with literature.

-line 99-101 – statement is unclear.

-line 11 – a period is needed at the end of the sentence.

-line 123-125 – in the paragraph, a description of generational diversity is provided but the relevance of this concept is not made explicit. Are you trying to argue that the unique needs of each cohort need to be considered? You need to be more explicit about how generational diversity is a problem that needs to be solved by administrators. You then make a statement about horizontal violence but have not described how horizontal violence and generational diversity are linked…this part of the discussion needs to be expanded.

-lines 130-134: The framework is not clearly presented in this paragraph. There are also several typos in this particular sentence.

-lines 135-136 – Sentence should not begin with ‘And’. Need to be more explicit about the link to a healthy work environment

-lines 137-146 – This discussion is not specific to generational diversity and seems to be inserted without context.

-p. 4

-lines 147-150 – The linkages made in this paragraph are too brief and need to be explained further. Suggest adding in a bridging statement to help with the transition to the next section on healthy work environments.

-line 152-54 – Statement unclear, suggest revising to “Healthy Work Environments have been described as workplaces that empower nurses to maintain personal satisfaction and achieve organizational objectives.”

-line 158 – suggest replacing ‘feature’ with ‘are characterized by’

-line 160-166 –These sentences are unclear (perhaps can be addressed by a better description of the framework as noted in comments above regarding page 3)

-line 166 – The discussion on generational diversity should be in a separate paragraph.

-line 171 – Suggest revising to, “…and healthy work environments for older nurses.”

-line 172-175 – the notion of a healing environment seems to be inserted without a full explanation, suggest deleting these sentences or rephrase

-line 176-79 – awkward sentence, suggest revising

-line 180-183 – the intent of this paragraph is unclear. Are you trying to say that horizontal violence leads to an unhealthy work environment which can then impact retention? Left unmanaged, can the consequences of horizontal violence be mitigated by other factors that support a healthy work environment (such as those listed by the AACN? Your argument needs to be more clear. Suggest combining the ideas in the next paragraph with this argument rather than 2 paragraphs.

-line 189 – Remove the extra words “is for the client to”.

-p5

-suggest making a first paragraph that highlights key findings about each content area

-begin the next paragraph with a  focus on your original argument i.e. the interaction of all components listed (nursing shortage, horizontal violence, generational diversity, healthy work environments) as per the title of the paper. Here you can mention the strategic  plan using the proposed framework

-the new material added regarding strategies needs to be positioned as reflecting these inter-relationships

-expand further on the notion of ‘multi-functionality’ that is mentioned in line 213

Author Response

Thank you for your insights.  

1)  Regarding the title: I have made the change that was suggested.

2)  Regarding page 1:  Anthropomorphism corrected.  The premises were in statement form.  The bulleted form for premises was maintained to focus attention. 

3)  Regarding page 2:  Suggestions addressed.  Additional researchers identified

4)  Regarding page 3:  Rewording and shortening of sentences were done to foster clarity areas reworded.  Note: the intent was not to present generational diversity as a problem-unto itself- but a phenomenon that needs to be considered. 

5)  Regarding page 4:   Sources added.  Rewording completed.  Extra words eliminated.

6)  Regarding page 5:  Rewording completed.  Multifunctionality addressed.